# Celiac disease and COVID-19 in adults: A systematic review

Parsa Amirian[1☯], Mahsa Zarpoosh[1☯*], Sajjad Moradi[2], Cyrus Jalili[3]

1 General Practitioner, Kermanshah University of Medical Science (KUMS), Kermanshah, Iran, 2 Nutritional Sciences Department, School of Nutritional Sciences and Food Technology, Kermanshah University of Medical Sciences, Kermanshah, Iran, 3 Medical Biology Research Centre, Health Technology Institute, Kermanshah University of Medical Sciences, Kermanshah, Iran

☯ These authors contributed equally to this work.
* zarpooshmahsa@gmail.com, mahsa.zarpoosh@kums.ac.ir

## Abstract

### Background

Celiac disease (CD) is an autoimmune disease affecting around 1.4% of the total human population. Local and systemic manifestations are described in CD. Viral infections seem to trigger CD or even have a worse outcome in CD patients. The evidence on the relationship between CD and coronavirus disease (COVID-19) is limited. To evaluate existing evidence on the association between CD and COVID-19, we conducted the current systematic review.

### Methods

We systematically searched Pubmed, Scopus, and Embase databases to find articles that reported risks or outcomes of COVID-19 in CD patients. Papers in any language published up to November 17, 2022, were evaluated for possible inclusion. The results were analyzed qualitatively. This study is registered with PROSPERO(CRD42022327380).

### Results

We identified 509 studies by searching databases; 14 reported data on the risk or outcome of COVID-19 in CD patients and were eligible for qualitative synthesis. We found that the relative risk of acquiring COVID-19 in CD patients may be lower than in the general population. Approximately 90% of infected patients were treated as an outpatient, and 10% were hospitalized. GFD adherence and Health-related quality of life (HR-QOL) were more or less the same before and during the pandemic. The gluten-free products (GFP) supply seems to be plunged during the pandemic. The data on the psychological effects of the pandemic were conflicting.

### Conclusion

The risk of acquiring COVID-19 in CD patients is lower than in the general population. Females were more likely to be infected by COVID-19, and the most common comorbidity in infected patients was a chronic lower respiratory disease; around 10% of infected patients

**Data Availability Statement:** All relevant data are within the paper and its Supporting Information files.

**Funding:** The authors received no specific funding for this work.

**Competing interests:** The authors have declared that no competing interests exist.

needed hospitalization, GFD adherence, and HR-QOL was more or less the same before and during the pandemic, depression, anxiety, and stress levels of patients varied among studies. Patients had more difficulties accessing GFPs based on limited data.

## Introduction

Affecting around 1.4% of the total human population, based on serologic tests, all over the globe, celiac disease (CD) is a common life-long disease with no curative treatment; the only successful therapy to contain the disorder is a strict gluten-free diet (GFD) [1]. Permanent diet restrictions, food supply chain issues, and lockdown limitations made bearing the coronavirus disease pandemic significantly harder for CD patients and reduced their health-related quality of life (HRQoL) [2, 3]. CD is an autoimmune disease caused by intolerance to peptide antigens by the immune system, originating from prolamins in wheat, rye, barley, and related grains [4]. Several local manifestations (e.g., villous atrophy, intestinal crypt hypertrophy, hikes in the number of lymphocytes in the epithelium and lamina propria, etc.) and systemic manifestations (e.g., iron deficiency anemia, osteoporosis, neurological diseases, etc.) have been reported in CD [5]. The CD incidence rate is growing quite rapidly; this trend cannot only be imputed to genetic background, environmental factors such as viral infections, namely reovirus, rotavirus, and enterovirus infections, can trigger CD; but the role of genetic background, especially human leukocyte antigen (HLA) DQ2 and DQ8 should not be neglected in the activation of T cell response to gluten peptides [5]. Coronavirus disease 2019 (COVID-19) is one the most prominent public health emergencies in recent memory, with more than 626 million confirmed cases and 6.5 million deaths, and various organ-specific manifestations, including manifestations in the gastrointestinal tract such as nausea, vomiting, and diarrhea [6, 7]. Despite the efforts, most CD patients perceive that they are at a higher risk or are clueless about the impact of COVID-19 infection on their health [8]. Therefore, to analyze the existing evidence of the risk and outcome of COVID-19 disease on CD patients, investigating levels of adhering to GFD (as the only accepted treatment for CD), food insecurity levels, depression, anxiety, stress levels in CD patients, and HRQoL of CD patients in the pandemic era, we conducted the current systematic review.

## Methods

All procedures used in the current systematic review have complied with the preferred reporting items for systematic reviews and meta-analysis guidelines [9]. This study is registered to PROSPERO (CRD42022327380).

### Search strategy

The MEDLINE/PubMed, SCOPUS, and EMBASE databases were systematically searched first time for published articles in any language until November 17, 2022. The search terms used were as follows: ("COVID-19"[Mesh] OR "COVID 19"[tiab] OR "SARS-CoV-2 Infection"[tiab] OR "2019 Novel Coronavirus Disease"[tiab] OR "2019-nCoV Infection "[tiab]) AND ("Celiac Disease"[Mesh] OR "Gluten Enteropathy"[tiab] OR "Gluten Enteropathies"[tiab] OR "Gluten Sensitive Enteropathy"[tiab] OR "Nontropical Sprue"[tiab] OR "Celiac Sprue"[tiab]). The detailed search terms used in each database are available in S1 File.

### Study selection

The inclusion criteria were as follows: (1) Original and peer-reviewed article, (2) studies with the adult population (>18 years old), (3) studies that report the risk of COVID-19 in CD patients by providing the number of total included CD patients, and the number of infected ones with COVID-19, or provide at least one outcome of COVID-19 such as hospitalization, intensive care unit (ICU) admission, death, or outpatient treatment in CD patients. The exclusion criteria were as follows: (1) reviews, short reports, letters, editorials, case reports, and conference abstracts, and (2) studies exclusively on the pediatric population (<18 years old) or pediatric majority population. We initially removed duplicate articles, then two authors (P.A., M.Z.) separately reviewed the titles and abstracts of the identified studies and excluded irrelevant studies. Full-text articles were then evaluated for potential inclusion, and the reference list of each full-text article was also assessed for additional qualified articles. Conflicting results were resolved via a second assessment with a third reviewer (C.J.).

### Data synthesis and quality assessment

The following data for each study were collected: first author, year of study publication, location of the study population, study design, number of total CD patients included in the study, number of CD patients infected with COVID-19, COVID-19 diagnosis method, sex, age, comorbid, GFD status before and during the pandemic, hospitalization, ICU admission, death, outpatient treatment, and key findings. Two authors conducted this procedure independently. The Newcastle-Ottawa Scale (NOS) [10] was used to assess the quality of selected studies in parallel by two reviewers (P.A., S.M.), with discrepancies resolved by a third reviewer (M.Z.). Finally, the results of this study were structured in a qualitative synthesis.

## Results

### Study characteristics

We identified 509 articles for screening through our systematic research of databases (PubMed, 75; Embase, 258; Scopus, 176;) (Fig 1). After removing duplicates and reviewing titles and abstracts, 41 studies underwent full-text analysis. In addition, 27 studies were excluded due to Inadequate information (n = 10), pediatric majority population (n = 6), and unoriginal articles (n = 11). Ultimately 14 studies were included in our systematic review [11–24]. All papers were observational studies; 12 articles assessed the risk of COVID-19 infection in CD patients [12–17, 19–24], nine articles evaluated the outcome of COVID-19 infection in CD patients [11, 14–18, 20, 21, 24], and eight articles provided information on CD patients GFD status [12, 13, 15–17, 21, 22, 24] characteristics of each study is summarized in Table 1.

### Quality of studies

The detailed quality assessment results of the studies are available in the S1 Table. Two studies had excellent quality, four studies had good quality, six studies had satisfactory quality, and two studies had unsatisfactory quality, according to NOS.

### Risk of COVID-19 in CD patients

Twelve studies have mentioned the total number of CD patients included in their studies; subsequently, 45411 CD patients are included in our review. Different methods, including polymerase chain reaction (PCR), COVID-19 antibodies, and computerized chest tomography (CT) scans, were used to diagnose COVID-19 in the studies. Four studies did not specify the diagnosis method of COVID-19 in their patients [13, 18, 20, 22]. Four studies reported

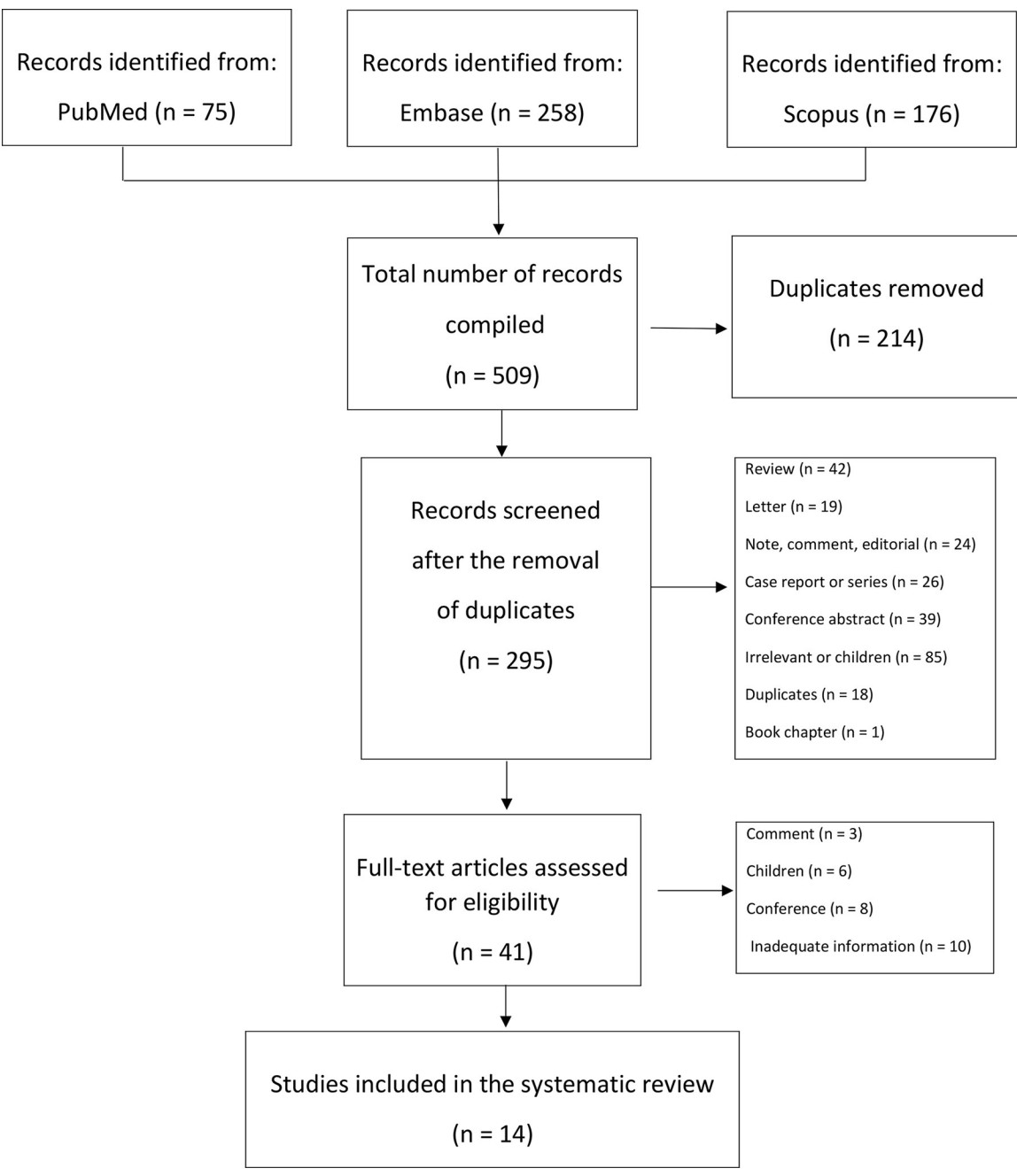

**Fig 1. PRISMA flow chart diagram.**

symptomatic (COVID-19/flu-like symptoms) patients and did not disclose positive lab tests in these patients [12, 17, 23, 24]. We identified 135 CD patients who were definitively diagnosed with COVID-19 (107 patients via PCR test, 31 patients via antibody test, two patients via chest CT scan, and 36 patients were diagnosed by two methods of diagnosis) by excluding CD populations, that did not report diagnosis based on methods mentioned above of diagnosis, 2595 CD patients were left. The pooled prevalence rate of COVID-19 infection in CD patients was 5.20%, according to our calculations; we also assessed the pooled relative risk of COVID-19

Table 1. Summary of included studies.

| Study/Year | Location | Study design | No. Total celiac | No. Celiac COVID-19 +, And basis of Dx | Male/Female | Age ± SD | Comorbidity | GFD adherence in CD patients | Outcome | health-related Quality of life in CD patients (HR-QOL) | Depression, anxiety, and stress levels in CD patients | Food insecurity | Data Risk | Data Outcome |
|---|---|---|---|---|---|---|---|---|---|---|---|---|---|---|
| Gökden. Y. 2020 | Istanbul, Turkey | Observational | 101 | 2 (PCR) | 0:2 (2F) | 34.5 (27y, 42y) | NM | Before: Completely = 66 Mostly = 25 Rarely = 7 Incompatible = 3 During: Completely = 59 Mostly = 33 Rarely = 6 Incompatible = 3 | Hospitalization: 1 ICU stay: 0 Death: 0 Out-patient: 1 | NM | STAI scale patients' state anxiety index = 40.7 ±7.9, the trait anxiety index = 44.5 ±8.5, (All patients were evaluated as mildly anxious) | Never had Difficulty in GFP Supply: Before = 36 (35.6%), During = 20 (19.8%). sometimes had difficulty in GFP supply: Before = 46 (45.5%), During = 39 (38.6%). always had difficulty in GFP supply: Before = 19 (18.8%), During = 42 (41.6%). | Yes | Yes |
| Al Hayek, A. 2020 | Riyadh, Saudi Arabia | Observational (retrospective) | NM | 6 (PCR) | NM | NM | T1DM: 6 | NM | Hospitalization: 2 ICU stay: NM Death: 0 Out-patient:4 | NM | NM | NM | No | Yes |
| Gasbarrini, G. 2021 | Multi center, Rome, Italy | Observational (retrospective cohort) | 542 | 5 (PCR) | NM | NM | NM | NM | Hospitalization: 0 ICU stay: 0 Death: 0 Out-patient: 5 | NM | NM | NM | Yes | Yes |
| Hadi, Y. B. 2022 | Multi center, USA | Observational (Retrospective cohort) | NM | 930 (NM) | 1:3.04 (230M,700F) | 46.50 ±18.10 | HTN: 323 Chronic lower respiratory disease: 331 T2DM: 179 IHD: 105 BMI>30: 263 Nicotine dependence: 97 | NM | Hospitalization: 81 ICU stay: 24 Death: 13 within 60 days Out-patient: 849 | NM | NM | NM | No | Yes |
| Lebwohl, B. 2021 | country-wide, Sweden | Observational (ESPRESSO cohort) | 40,963 | 414 (NM) | NM | NM | NM | NM | Hospitalization: 58 ICU stay: NM Death: 22 (Overall mortality rate); 11 (among hospitalized) Out-patient: NM Severe COVID-19: 24 | NM | NM | NM | Yes | Yes |
| Falcomer, A. L. 2021 | country-wide, Brazil | Observational (cross-section) | 674 | 42 (positive test) | NM | NM | NM | During Always follow = 597 Not always follow = 77 | NM | CDQ questionnaire (Range, 28–196) mean ± SD 125.26 ± 32.02 | NM | NM | Yes | No |

(Continued)

**Table 1.** (Continued)

| Study/Year | Location | Study design | No. Total celiac | No. Celiac COVID-19 +, And basis of Dx | Male/Female | Age ± SD | Comorbidity | GFD adherence in CD patients | Outcome | health-related Quality of life in CD patients (HR-QOL) | Depression, anxiety, and stress levels in CD patients | Food insecurity | Data Risk | Data Outcome |
|---|---|---|---|---|---|---|---|---|---|---|---|---|---|---|
| Elli, L. 2021 | Milan, Italy | Observational (prospective) | 362 | 42 (COVID-19-like symptom) | 1:6 (6M, 36F) | 45±15 | 19* (Auto immune) | 10* Non-adherent to a GFD and/or positive urinary GIP None reported difficulty adhering to a GFD (NRS 9.7 ± 0.9) *** | Hospitalization: 1 ICU stay: 0 Death: 0 Out-patient: 41 | NM | ISMA stress questionnaire: low probability of having stress-related illness = 12% (0–4) stress-related problems = 73% (5–13) higher probability of having stress-related problems = 15% (14–25) | NM | Yes | No |
| | | | | 20 (Anti-SARS-CoV-2 Ig): 16(IgA) 15(IgG) 15(anti RBD) 13(anti N) 1 (PCR) | 1:4 (4M, 16F) | 47±13 | 2 (Auto immune) | | | | | | | |
| Greco, N. 2022 | Rome, Italy | Observational | 191 | 11 positive tests: 9(PCR) 2(Antibody test) | NM | NM | NM | NM | Hospitalization: 0 ICU stay: 0 Death: 0 Out-patient: 11 | NM | NM | 19 (9.9%): had difficulty searching for specific food 11 (5.8%): had difficulty managing diet | Yes | Yes |
| Li, J. 2022 | UK | Observational (UK Biobank cohort) | 535 | 74 (PCR) | NM | NM | NM | NM | Severe COVID-19: 3 Non-severe COVID-19: 71 | NM | NM | NM | Yes | Yes |
| Schiepatti, A. 2021 | Pavia, Italy | Observational (cohort) | 324 | 9 positive tests 5(PCR) 6 (Serology) | 1:3.5 (2M, 7F) | 35±12 | Breast cancer: 1 Epilepsy: 1 Bronchial asthma: 1 Smoke: 1 | During Adherent to GFD = 306 Non-adherent to GFD = 10 Not available = 8 | Hospitalization: 0 ICU stay: 0 Death: 0 Out-patient: 9 | NM | NM | NM | Yes | Yes |
| | | | | 44 (COVID-19-like symptoms) | 1:3 (11M, 33F) | 41±17 | NM | | NM | | | | | |
| Ibsen, J. H. 2022 | Oslo, Norway | Observational (Case-control) | 85 | 3 (Ab>10 BAU/ml Pre-vaccination) | NM | NM | NM | NM | NM | NM | NM | NM | Yes | No |
| Mehtab, W. 2021 | country-wide, India | Observational | 505 | 25 (NM) | NM | NM | NM | Before: (Mean CDAT = 12.08 ±3.3) Good = 296 (58.6%) (CDAT score <13) Average = 175 (34.6%) (CDAT score 13–17) Poor = 34 (6.7%) (CDAT score >17) During: (Mean CDAT = 12.37 ±3.6) Good = 263 (52.1%) (CDAT score <13) Average = 178 (35.2%) (CDAT score 13–17) Poor = 64 (12.6%) (CDAT score >17) | NM | median CD-QOL score of patients during lockdown = 56 (Range: 20–93) | NM | heavy delivery charges for getting GFP at home = 265 (52.4%) long distance to procure GFP = 227 (44.9%) no transport available to get GFP = 112 (22.1%) Increased prices of GFP = 218 (43.1%) shortage of money to buy GFP = 105 (20.7%) shortage of grains = 141 (27.9%) disrupted courier services = 25 (4.9%) | Yes | No |

*(Continued)*

**Table 1.** (Continued)

| Study/Year | Location | Study design | No. Total celiac | No. Celiac COVID-19 +, And basis of Dx | Male/Female | Age ± SD | Comorbidity | GFD adherence in CD patients | Outcome | health-related Quality of life in CD patients (HR-QOL) | Depression, anxiety, and stress levels in CD patients | Food insecurity | Data Risk | Data Outcome |
|---|---|---|---|---|---|---|---|---|---|---|---|---|---|---|
| Gholam-Mostafaei, F. S. 2022 | Tehran, Iran | Observational (cross-section) | 455 | 11 (PCR) or 2 (CT scan) | 1:2.66 (3M, 8F) | NM | (Over weight: 2 Obese: 1 BMI >25: 4) * Smoke: 1 T2DM: 2 Cardiovascular disease: 1 Thyroid dysfunction: 1 Liver disorder: 1 Neurological disorder: 1** | NM | Hospitalization: 1 ICU stay: 0 Death: 0 Out-patient: 10 | NM | NM | NM | Yes | Yes |
| Möller, S. P. 2021 | multi-center (Australia, New Zealand &…) | Observational (cohort) | 674 | 3 (having COVID-19 symptoms) | NM | NM | NM | Before: Adequate adherence = 648 (63.3%) (CDAT score <13) Inadequate adherence = 375 (36.7%) (CDAT score ≥13) During: Adequate adherence = 453 (67.2%) (CDAT score <13) Inadequate adherence = 221 (32.8%) (CDAT score ≥13) | NM | EUROHIS-QOL: (Range, 8–40) (mean ±SD) Before 32.35 ±5.34 during 32.70 ±5.06 (Higher scores = greater QOL) | Depression, anxiety, stress scales (DASS-21) (Range: 0–126) mean ±SD Before = 18.95 ±18.35 After = 18.07 ±17.53 | NM | Yes | No |

Abbreviation: ICU intensive care unit, NM not mentioned.

*Conflicting data and potentially wrong

**Some patients had two or more comorbidities

*** Numerical rate scale (NRS) from 0 to 10

infection acquisition in CD patients compared to the general population using the data provided by the world health organization (WHO) [25]. The pooled relative risk was 0.63 (95% CI, 0.54–0.75). The total number of CD patients reported to be infected by COVID-19, regardless of the method of diagnosis (some patients only had COVID-19 symptoms), was 705 out of 45411 CD patients included in this review. The pooled prevalence rate of COVID-19 infection in CD patients was 1.55%, and the pooled relative risk of COVID-19 infection acquisition was 0.19 (95% CI, 0.17–0.20).

## Sex, age, and comorbidities of CD patients infected with COVID-19

The total number of COVID-19-infected CD patients (symptomatic/laboratory confirmed) in 14 studies was 1641 patients; five studies have mentioned patients' gender, 802 (48.87%) patients were females, 256 (15.60%) were males, and gender of 583 (35.52%) patients were not mentioned [12, 15, 16, 18, 24]. Four studies mentioned patients' age; the mean age ranged from 34 to 47 years [12, 16, 18, 24]. Two studies separately reported the mean age of the laboratory-confirmed and symptom-only groups [12, 24]. Five studies reported the comorbidities of COVID-19-infected patients [11, 12, 15, 18, 24]. We spotted 1340 comorbidities in these studies, and they were as follows: type 1 diabetes mellitus (6(0.44%)), hypertension(323(24.10%)), chronic lower respiratory disease(331(24.70%)), type 2 diabetes mellitus (181(13.50%)), ischemic heart disease (105(7.83%)), overweight/obese (267(19.92%)), nicotine dependence/smoking (99(7.38%)), autoimmune diseases (21(1.56%)), breast cancer (1(0.07%)), epilepsy(1 (0.07%)), asthma(1(0.07%)), cardiovascular disease(1(0.07%)), thyroid dysfunction (1(0.07%)), liver disorder (1(0.07%)), and neurological disorder (1(0.07%)); some patients may have more than one comorbidity.

## Outcomes of CD patients infected with COVID-19

Ten studies reported outcomes of CD patients infected with COVID-19 [11, 12, 14–18, 20, 21, 24]. One study reported outcomes of the symptomatic-only group (12), and another reported outcomes of PCR-confirmed patients [24]. Nine hospitalized patients were reported in 9 studies [11, 12, 14–18, 20, 24]; out of 1430 patients, 144 (10.06%) were hospitalized. Seven studies stated the number of intensive care unit (ICU) admitted patients; Hadi et al. reported that 24 (2.58%) patients out of 930 patients recruited in their study needed ICU admission; other studies reported that none of the study patients were admitted to ICU [12, 14–18, 24]. Nine studies provide information on whether CD patients lost their lives due to COVID-19 or not; Hadi et al. reported that, In the 30-day and 60-day periods after COVID-19 infection, 12 and 13 deaths, respectively, occurred in CD patients, Lebwohl et al. reported that the overall mortality rate in CD patients was 5.3%, and 11 patients died after hospitalization; other studies reported no deaths as the result of COVID-19 [11, 12, 14–18, 20, 24]. Eight studies provided data on out-patient treatment; 930 (91.53%) patients received out-patient treatment out of 1016 [11, 12, 14–18, 24]. Additionally, two studies reported outcomes in terms of severe and non-severe COVID-19 infected groups; severely infected was defined as COVID-19-related death and ICU admission. Li et al. found 3 (4.05%) patients were in the severely infected COVID-19 group, and 71 (95.94%) patients were in the non-severe group; in Lebwohl et al. study, 24 (5.79%) patients were severely infected by COVID-19 [20, 21].

## GFD adherence in CD patients before and during the pandemic

Six studies provided data on the GFD adherence status of CD patients before or during the pandemic [12, 13, 16, 22–24]. Three articles presented data comparing GFD adherence in CD patients before and during the pandemic [16, 22, 23]. Gökden et al. found that before the

pandemic, 66 (65.34%) patients adhered to GFD completely, 25 (24.75%) patients mostly stuck, 7 (6.93%) patients rarely attached, and 3 (2.97%) patients were incompatible; the results during the pandemic were as follows respectively: 59 (58.41%), 33 (32.67%), 6 (5.94%), 3 (2.97%) [16]. Other two studies used the CD adherence test (CDAT) to compare adherence to GFD before and during the pandemic; Mehtab et al. found before the pandemic, 296 (58.6%) patients had a good GFD compliance (CDAT score < 13), 175 (34.6%) patients had average GFD compliance (CDAT score 13–17), and 34 (6.7%) patients had poor GFD compliance (CDAT score >17); during the pandemic 263 (52.1%) patient had a good GFD compliance (CDAT score < 13), 178 (35.2%) patients had average GFD compliance (CDAT score 13–17), and 64(12.6%) patients had poor GFD compliance (CDAT score >17) [22]. Möller et al. found that before the pandemic, 648 (63.3%) patients had adequate adherence (CDAT score < 13), and 375 (36.7%) patients had inadequate adherence (CDAT score ≥ 13). During the pandemic, 453 (67.2%) patients had adequate compliance (CDAT score < 13), and 221 (32.8%) patients had inadequate adherence (CDAT score ≥ 13) [23]. The other three studies only reported GFD adherence during the pandemic [12, 13, 24]. Out of 674 patients recruited in the Falcomer et al. study during the pandemic, 597 (88.57%) patients always followed GFD, and 77 (11.42%) patients did not always follow GFD; in the Schiepatti et al. study, out of 324 patients 306 (94.44%) adhered to GFD during the pandemic, 10 (3.08%) patients did not adhere to GFD, and data was unavailable in 8 (2.46%) patients [13, 24]. Elli et al. stated that none of the included patients reported difficulty adhering to GFD, and 10 (2.76%) patients were non-adherent to GFD, or gluten immunogenic peptides (GIP) were detectable in their samples [12].

## Health-related quality of life (HR-QOL) in CD patients

Falcomer et al. used the celiac disease quality of life questionnaire (CDQ) to assess the QOL of CD patients during the pandemic; the questionnaire comprises four subgroups, including emotion, social, worries, and gastrointestinal; it ranges from 28 to 196. The mean score ± standard deviation among their population study was 125.26 ± 32.02; a higher score means better QOL; they found highly educated patients had better QOL, and no significant difference was seen in QOL between patients infected by COVID-19 or not infected by it [13]. Mehtab et al. used CD-related quality of life (CD-QOL) questionnaire; it comprises four sub-scales, namely limitations, dysphoria, health concerns, and inadequate treatment; it ranges between 20–93. The median score of their study population was 56; they reported that approximately 45% of patients had high CD-QOL scores [22]. Lastly, Möller et al. used the EURO-HIS-QOL questionnaire before and during the pandemic to assess changes in the QOL of CD patients ranging between 8 to 40. The mean score ± standard deviation before the pandemic was 32.35 ± 5.34, and during the pandemic was 32.70 ± 5.06 [23, 26].

## Depression, anxiety, and stress levels in CD patients

Gökden et al. used the State-trait Anxiety Inventory (STAI) scale to evaluate state anxiety and trait anxiety levels in CD patients during the pandemic; it ranges from 20 to 80, and higher scores are associated with higher levels of stress; patients' mean ± SD state anxiety score was 40.7 ± 7.9, and their mean ± SD trait anxiety score was 44.5 ± 8.5; authors also reported that all patients were evaluated as mildly anxious [16, 27]. Elli et al. used the International Stress Management Association (ISMA) Stress questionnaire to assess the stress levels of CD patients, it ranges from 0 to 25, and higher scores are correlated with higher stress levels; 12% of patients had a low probability of having a stress-related illness (scored, 0–4), 73% had stress-related problems (achieved, 5–13), and 15% had a higher probability of having stress-related problems

(scored, 14–25) [12]. Möller et al. used depression anxiety stress scale-21 (DASS21) to estimate psychological distress in CD patients before and during the pandemic, it ranges from 0 to 126, and higher scores demonstrate higher levels of psychological distress; before the pandemic, the mean ± SD was 18.95 ± 18.35, and during the pandemic was 18.07 ± 17.53 [23, 28].

## Food insecurity in CD patients before and during the pandemic

Gökden et al. compared gluten-free products (GFP) supply difficulties before and during the pandemic; before the pandemic, 36 (35.6%) patients never had trouble with GFP supply, 46 (45.5%) sometimes had difficulty in GFP supply, and 19 (18.8%) always had a problem in GFP supply; during the pandemic, 20 (19.8%) patients never had difficulty in GFP supply, 39 (38.6%) sometimes had difficulty in GFP supply, and 42 (41.6%) always had difficulty in GFP supply [16]. Greco et al. also provided some information on GFP supply difficulties and reported 19 (9.9%) patients had difficulty searching for specific food; additionally, 11 (5.8%) patients had difficulty managing their diet [17]. Mehtab et al. indicated main GFP supply difficulties during lockdowns, 265 (52.4%) patients faced serious delivery charges for getting GFP at home, 227 (44.9%) complained about the long distance to procure GFP, 112 (22.1%) complained that there was no available transportation to get GFP, 218 (43.1%) criticized increased prices of GFP, 105 (20.7%) faced a shortage of money with buying GFP, 141 (27.9%) faced a shortage of grains, and 25 (4.9%) criticized disrupted courier services [22].

## Discussion

In this systematic review, we included 14 original articles with adult majority populations; risk and outcome of COVID-19 in CD in patients, patient characteristics, GFD adherence before and during the pandemic, HR-QOL, depression, anxiety, stress levels, GFP availability, and food insecurity was discussed. Whenever authors provided pre-pandemic and mid-pandemic data, we compared the results to have a holistic view of the impact of the pandemic. Our findings suggest that CD patients are at lower risk of COVID-19 infection acquisition, the relative risk using laboratory-confirmed patient's subgroup was 0.63 (95% CI, 0.54–0.75), and it was 0.19 (95% CI, 0.17–0.20) when we used all patient's data regardless the diagnosis method. Normally, the relative risk of disease should be higher when symptomatic-only patients are included; in our calculations, it was quite the opposite, which suggests accumulated misdiagnosis through articles reporting other methods of diagnosis rather than definite ones. CD patients may experience more subclinical forms of COVID-19 infection, which can justify their lower relative risk. To the best of our knowledge, this hypothesis has not been tested. Several factors may play a role in lower COVID-19 infection acquisition rates of CD patients than the general population [25]. Generally, the incidence rate of CD in women is two times more than in men [29]. Still, in our study, female CD patients were three times more likely to be infected by COVID-19 than male CD patients, which makes the female gender a predisposing factor to COVID-19 in CD patients. Fifteen different comorbidities were mentioned in the included studies; the top three most common ones (chronic lower respiratory diseases, hypertension, obesity) account for roughly 70% of the comorbidities, suggesting that CD patients with mentioned comorbidities may be at higher risk for COVID-19 infection acquisition. We recommend CD patients with these comorbidities take extra caution during the pandemic. In order to justify the lower risk of COVID-19 acquisition in CD patients, two main factors can play a role: environmental and genetic factors. CD patients may follow protective measures against COVID-19 more commonly, for instance, due to visiting their doctors frequently and having a better patient-centered care or because of the concern of being more susceptible to the COVID-19 infection. CD is an autoimmune disease with a robust genetic background;

almost all CD patients carry one of two human leukocyte antigens (HLA-DQ2 or DQ-8) [5]. The role of HLA molecules in the susceptibility and severity of viral infections has been investigated before; HLA-DQ2 or DQ-8 may have a protective effect against the SARS-CoV-2 virus; Tavasolian et al. investigated the association between genetic susceptibilities and immune response to the novel coronavirus infection and found that the COVID-19 infection demographic may be related to HLA profiles of the region [17, 30]. Langton et al. investigated the association between the HLA genotype and the severity of COVID-19 infection. They found that HLA-DRB1*04:01 is significantly more frequent in severe COVID-19 patients compared to the asymptomatic staff group; they found thatHLA-DQA1*01:01, HLA-DQB1*05:01, and HLA-DRB1*01:01 are less frequent in the asymptomatic group compared to the background population [31]. Almost 10% of infected patients were hospitalized, and 90% did not need hospitalization. Although Gökden et al. compared of GFP accessibility before and during the pandemic showed patients had more difficulty accessing GFPs, other studies showed that GFD adherence did not change drastically during the pandemic [16, 22, 23]. Articles reported contradictory data on depression, anxiety, and stress levels, but all concluded that CD patients maintained a good HR-QOL during the pandemic [12, 13, 16, 22, 23]. As the end, COVID-19 pandemic looms, and the endemic phase of the disease begins [32], the necessity of understanding the risks and outcomes of COVID-19 and CD co-presence when exposure to COVID-19 is in childhood becomes clearer. According to our latest information, this is the first systematic review that evaluates the impact of the pandemic on CD patients. We evaluated different domains with regard to CD patients in the pandemic era to have a holistic view of the situation of CD patients during the COVID-19 pandemic; we created two subgroups to calculate the relative risk of COVID-19 acquisition and to achieve strong causal evidence of the subject we excluded short reports, letters, editorials, case reports, and conference abstracts, 45411 CD patients were included in our study, which has not been done before on this subject. We faced several limitations in conducting this systematic review; firstly, all studies were observational studies, none were a randomized clinical trial, many used questionnaires, and were done through social media or telephone; these study designs are not among the best designs, through our quality assessment process we found that almost half of the studies' total quality was satisfactory. Second, excluding one or two studies, others had small sample sizes, and the data on COVID-19-infected patients were not reported properly. We found conflicting data in the two included studies. Third, we excluded case reports and studies that only reported a single patient due to insufficient convincing results and their excessive number. Fourth, we excluded studies with pediatric-only or pediatric majority populations because of different manifestations of both CD and COVID-19 in children; we believe that a different systematic review is needed for pediatric populations. In conclusion risk of COVID-19 infection is lower in CD patients than in the general population, females were more likely to be infected by COVID-19, and the most common comorbidity in infected patients was a chronic lower respiratory disease; around 10% of infected patients needed hospitalization, GFD adherence, and HR-QOL was more or less the same before and during the pandemic, depression, anxiety, and stress levels of patients varied among studies. Patients had more difficulties accessing GFPs based on limited data.

## Supporting information

**S1 File. Search strategy.**
(DOCX)

**S2 File. Prisma flow chart.**
(DOC)

**S1 Table. Quality assessment.**
(DOCX)

## Author Contributions

**Data curation:** Parsa Amirian, Mahsa Zarpoosh, Sajjad Moradi.

**Formal analysis:** Parsa Amirian.

**Investigation:** Cyrus Jalili.

**Methodology:** Parsa Amirian.

**Supervision:** Sajjad Moradi, Cyrus Jalili.

**Validation:** Sajjad Moradi, Cyrus Jalili.

**Visualization:** Mahsa Zarpoosh, Sajjad Moradi.

**Writing – original draft:** Parsa Amirian, Mahsa Zarpoosh.

**Writing – review & editing:** Parsa Amirian, Mahsa Zarpoosh.

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
