## [Decision Letter · Decision Letter 0]

28 Mar 2023

PONE-D-23-03782Celiac Disease and COVID-19 in Adults: A Systematic ReviewPLOS ONE

Dear Dr. Zarpoosh,

Thank you for submitting your manuscript to PLOS ONE. After careful consideration, we feel that it has merit but does not fully meet PLOS ONE’s publication criteria as it currently stands. Therefore, we invite you to submit a revised version of the manuscript that addresses the points raised during the review process.

We look forward to receiving your revised manuscript.

Kind regards,

María de Lourdes Moreno Amador, Ph.D.

Academic Editor

PLOS ONE

Journal Requirements:

3. We note that this manuscript is a systematic review or meta-analysis; our author guidelines therefore require that you use PRISMA guidance to help improve reporting quality of this type of study. Please upload copies of the completed PRISMA checklist as Supporting Information with a file name “PRISMA checklist”.

Additional Editor Comments:

This manuscript assesses a systematic review on COVID-19 in adults with celiac disease. The topic studied is relevant for clinical practice in gastroenterology, general medicine and in the discovery of predisposing factors to the virus. I recommend the major revision of the manuscript, due mainly to the temporary uncluttered study design as well as the certain poor, imprecise sentences provided in the manuscript. The main arguments are listed below:

1. Since the review is based on the COVID 19 pandemic that appeared in December 2019, the authors should study articles after December 2019, not before December 2019. Therefore, many of the studies referred to prior to this date should be eliminated:

2. The authors assert that the risk of acquiring COVID-19 in CD patients is lower than in the general population but do not explain in detail. Moreover, in patients in whom you found a risk, they also do not detail exactly why.

3.What's new in your article? The importance of this article in relation to existing articles should be emphasized, describing what is new about it.

4. Important findings should be reflected in the abstract "females were more likely to be infected by COVID-19, and the most common comorbidity in infected patients was a chronic lower respiratory disease; around 10% of infected patients needed hospitalization, GFD adherence, and HR-QOL was more or less the same before and during the pandemic, depression, anxiety, and stress levels of patients varied among studies. Patients had more difficulties accessing GFPs based on limited data.”

5. There are incoherent sentences in the current situation, so it is recommended to eliminate them and try to show own conclusions: "Further studies with larger populations are needed" or "Further studies with more robust designs are needed to draw definite conclusions".

Reviewers' comments:

Reviewer's Responses to Questions

**Comments to the Author**

1. Is the manuscript technically sound, and do the data support the conclusions?

Reviewer #1: Partly

2. Has the statistical analysis been performed appropriately and rigorously? 

Reviewer #1: Yes

3. Have the authors made all data underlying the findings in their manuscript fully available?

Reviewer #1: Yes

4. Is the manuscript presented in an intelligible fashion and written in standard English?

Reviewer #1: Yes

5. Review Comments to the Author

Reviewer #1: 1.The COVID 19 pandemic appeared in December 2019. If you want to study the articles on this topic, you study articles after December 2019, not before December 2019:

1.Singh P, Arora A, Strand TA, Leffler DA, Catassi C, Green PH, Kelly CP, Ahuja V,

Makharia GK. Global prevalence of celiac disease: systematic review and meta-analysis.

Clinical gastroenterology and hepatology. 2018 Jun 1;16(6):823-36.

2. Bascuñán KA, Vespa MC, Araya M. Celiac disease: understanding the gluten-free diet.

European Journal of Nutrition. 2017 Mar;56(2):449-59.

5. Shan L, Molberg Ø, Parrot I, Hausch F, Filiz F, Gray GM, Sollid LM, Khosla C. Structural

basis for gluten intolerance in celiac sprue. Science. 2002 Sep 27;297(5590):2275-9. doi:

10.1126/science.1074129. PMID: 12351792.

6. Rodrigo L. Celiac disease. World J Gastroenterol. 2006 Nov 7;12(41):6577–84. doi:

10.3748/wjg.v12.i41.6585. Epub 2006 Nov 7. PMCID: PMC4125661.

7. Bouziat R, Hinterleitner R, Brown JJ, Stencel-Baerenwald JE, Ikizler M, Mayassi T,

Meisel M, Kim SM, Discepolo V, Pruijssers AJ, Ernest JD, Iskarpatyoti JA, Costes LM,

Lawrence I, Palanski BA, Varma M, Zurenski MA, Khomandiak S, McAllister N,

Aravamudhan P, Boehme KW, Hu F, Samsom JN, Reinecker HC, Kupfer SS, Guandalini S,

Semrad CE, Abadie V, Khosla C, Barreiro LB, Xavier RJ, Ng A, Dermody TS, Jabri B.

Reovirus infection triggers inflammatory responses to dietary antigens and development of

celiac disease. Science. 2017 Apr 7;356(6333):44-50. doi: 10.1126/science.aah5298. PMID:

28386004; PMCID: PMC5506690

9. Mårild K, Fredlund H, Ludvigsson JF. Increased risk of hospital admission for influenza in

patients with celiac disease: a nationwide cohort study in Sweden. Am J Gastroenterol. 2010

Nov;105(11):2465-73. doi: 10.1038/ajg.2010.352. Epub 2010 Sep 7. PMID: 20823839.

10. Dieterich W, Esslinger B, Schuppan D. Pathomechanisms in celiac disease. International

archives of allergy and immunology. 2003;132(2):98-108.

31. S. Schmidt, H. Mühlan, M. Power, The EUROHIS-QOL 8-item index: psychometric

results of a cross-cultural field study, Eur. J. Pub. Health 16 (4) (2005) 420–428.

33. S.H. Lovibond, P.F. Lovibond, Manual for the Depression Anxiety Stress Scales, 2nd.

Ed., Psychology Foundation, Sydney, 1995. ISBN: 7334-1423-0.

34. Vader W, Stepniak D, Kooy Y, Mearin L, Thompson A, van Rood JJ, Spaenij L, Koning

F. The HLA-DQ2 gene dose effect in celiac disease is directly related to the magnitude and

breadth of gluten-specific T cell responses. Proceedings of the National Academy of

Sciences. 2003 Oct 14;100(21):12390-5.

35. Dendrou CA, Petersen J, Rossjohn J, Fugger L. HLA variation and disease. Nature

Reviews Immunology. 2018 May;18(5):325-39.

36. Lin M, Tseng HK, Trejaut JA, Lee HL, Loo JH, Chu CC, Chen PJ, Su YW, Lim KH, Tsai

ZU, Lin RY. Association of HLA class I with severe acute respiratory syndrome coronavirus

infection. BMC medical genetics. 2003 Dec;4(1):1-7.

THE ARTICLES ABOVE DO NOT CORRESPOND WITH THE TITLE OF THE ARTICLE

2."The risk of acquiring COVID-19 in CD patients is lower than in the general population"

The risk is low! ok, but develop a bit. For the patients in whom you found a risk, detail it! What is it about? Why did they contact the virus?

3.What's new in your article?

4.PUT THESE CONCLUSIONS IN ABSTRACT: "females were more likely to be infected by COVID-19, and the

most common comorbidity in infected patients was a chronic lower respiratory disease;

around 10% of infected patients needed hospitalization, GFD adherence, and HR-QOL was

more or less the same before and during the pandemic, depression, anxiety, and stress levels

of patients varied among studies. Patients had more difficulties accessing GFPs based on

limited data

5.DELETE THIS:

DELETE: "Further studies with larger populations are needed." BECAUSE The COVID19 pandemic is no more. If you ask for this, does it mean you want another COVID19 pandemic to come?

DELETE: "Further studies with more robust designs are needed to draw definite conclusions" - From what you have, show some conclusions!

6. PLOS authors have the option to publish the peer review history of their article (what does this mean?). If published, this will include your full peer review and any attached files.

Reviewer #1: **Yes: **SAMASCA GABRIEL

---

## [Author Response · Author response to Decision Letter 0]

31 Mar 2023

Reviewer #1: 1.The COVID 19 pandemic appeared in December 2019. If you want to study the articles on this topic, you study articles after December 2019, not before December 2019:

1.Singh P, Arora A, Strand TA, Leffler DA, Catassi C, Green PH, Kelly CP, Ahuja V,

Makharia GK. Global prevalence of celiac disease: systematic review and meta-analysis.

Clinical gastroenterology and hepatology. 2018 Jun 1;16(6):823-36.

2. Bascuñán KA, Vespa MC, Araya M. Celiac disease: understanding the gluten-free diet.

European Journal of Nutrition. 2017 Mar;56(2):449-59.

5. Shan L, Molberg Ø, Parrot I, Hausch F, Filiz F, Gray GM, Sollid LM, Khosla C. Structural

basis for gluten intolerance in celiac sprue. Science. 2002 Sep 27;297(5590):2275-9. doi:

10.1126/science.1074129. PMID: 12351792.

6. Rodrigo L. Celiac disease. World J Gastroenterol. 2006 Nov 7;12(41):6577–84. doi:

10.3748/wjg.v12.i41.6585. Epub 2006 Nov 7. PMCID: PMC4125661.

7. Bouziat R, Hinterleitner R, Brown JJ, Stencel-Baerenwald JE, Ikizler M, Mayassi T,

Meisel M, Kim SM, Discepolo V, Pruijssers AJ, Ernest JD, Iskarpatyoti JA, Costes LM,

Lawrence I, Palanski BA, Varma M, Zurenski MA, Khomandiak S, McAllister N,

Aravamudhan P, Boehme KW, Hu F, Samsom JN, Reinecker HC, Kupfer SS, Guandalini S,

Semrad CE, Abadie V, Khosla C, Barreiro LB, Xavier RJ, Ng A, Dermody TS, Jabri B.

Reovirus infection triggers inflammatory responses to dietary antigens and development of

celiac disease. Science. 2017 Apr 7;356(6333):44-50. doi: 10.1126/science.aah5298. PMID:

28386004; PMCID: PMC5506690

9. Mårild K, Fredlund H, Ludvigsson JF. Increased risk of hospital admission for influenza in

patients with celiac disease: a nationwide cohort study in Sweden. Am J Gastroenterol. 2010

Nov;105(11):2465-73. doi: 10.1038/ajg.2010.352. Epub 2010 Sep 7. PMID: 20823839.

10. Dieterich W, Esslinger B, Schuppan D. Pathomechanisms in celiac disease. International

archives of allergy and immunology. 2003;132(2):98-108.

31. S. Schmidt, H. Mühlan, M. Power, The EUROHIS-QOL 8-item index: psychometric

results of a cross-cultural field study, Eur. J. Pub. Health 16 (4) (2005) 420–428.

33. S.H. Lovibond, P.F. Lovibond, Manual for the Depression Anxiety Stress Scales, 2nd.

Ed., Psychology Foundation, Sydney, 1995. ISBN: 7334-1423-0.

34. Vader W, Stepniak D, Kooy Y, Mearin L, Thompson A, van Rood JJ, Spaenij L, Koning

F. The HLA-DQ2 gene dose effect in celiac disease is directly related to the magnitude and

breadth of gluten-specific T cell responses. Proceedings of the National Academy of

Sciences. 2003 Oct 14;100(21):12390-5.

35. Dendrou CA, Petersen J, Rossjohn J, Fugger L. HLA variation and disease. Nature

Reviews Immunology. 2018 May;18(5):325-39.

36. Lin M, Tseng HK, Trejaut JA, Lee HL, Loo JH, Chu CC, Chen PJ, Su YW, Lim KH, Tsai

ZU, Lin RY. Association of HLA class I with severe acute respiratory syndrome coronavirus

infection. BMC medical genetics. 2003 Dec;4(1):1-7.

THE ARTICLES ABOVE DO NOT CORRESPOND WITH THE TITLE OF THE ARTICLE

Response: Thank you for your comment. All references mentioned above were eliminated and were replaced with new references below (sometimes two or more references were replaced with one new and comprehensive reference). A few sentences about the accelerating celiac disease incidence rate from new reference #5 were added (Page 3. Line 2-4). Tavasolian et al. study conclusion (COVID-19 infection demographic may be related to HLA profiles of the region) was also added (Page 13, Line 19-21)

1. Lebwohl B, Rubio-Tapia A. Epidemiology, presentation, and diagnosis of celiac disease. Gastroenterology. 2021 Jan 1;160(1):63-75.

4. Sharma N, Bhatia S, Chunduri V, Kaur S, Sharma S, Kapoor P, Kumari A, Garg M. Pathogenesis of celiac disease and other gluten related disorders in wheat and strategies for mitigating them. Frontiers in Nutrition. 2020 Feb 7;7:6.

5. Iversen R, Sollid LM. The immunobiology and pathogenesis of celiac disease. Annual Review of Pathology: Mechanisms of Disease. 2023 Jan 24;18:47-70.

26. S.P. M¨oller, P. Apputhurai, J.A. Tye-Din, S.R. Knowles, Quality of life in coeliac disease: relationship between psychosocial processes and quality of life in a sample of 1,697 adults living with coeliac disease, J. Psychosom. Res. 151 (2021), 110652

28. Ali AM, Alkhamees AA, Hori H, Kim Y, Kunugi H. The Depression Anxiety Stress Scale 21: Development and Validation of the Depression Anxiety Stress Scale 8-Item in Psychiatric Patients and the General Public for Easier Mental Health Measurement in a Post COVID-19 World. Int J Environ Res Public Health. 2021 Sep 27;18(19):10142. doi: 10.3390/ijerph181910142. PMID: 34639443; PMCID: PMC8507889. 

29. King JA, Jeong J, Underwood FE, et al. Incidence of celiac disease is increasing over time: a systematic review and meta-analysis. Am J Gastroenterol 2020; 115:507–525.

30. Tavasolian F, Rashidi M, Hatam GR, Jeddi M, Hosseini AZ, Mosawi SH, Abdollahi E, Inman RD. HLA, immune response, and susceptibility to COVID-19. Frontiers in immunology. 2021 Jan 8;11:601886. 

2."The risk of acquiring COVID-19 in CD patients is lower than in the general population"

The risk is low! ok, but develop a bit. For the patients in whom you found a risk, detail it! What is it about? Why did they contact the virus?

Response: Thank you for your comment. Several reasons for lower relative risk in celiac disease patients were suggested, 1. due to misdiagnosis of COVID-19 infection in patients (Page 12, Line 39-42). 2. Celiac disease patients may experience more subclinical forms of COVID-19 infection (Page 12, Line 42,43; Page 13, Line 1). 3. Environmental Factors of lower risk namely: CD patients may follow protective measures against COVID-19 more commonly and having better patient-centered care or because of the concern of being more susceptible to the COVID-19 infection (Page 13, Line 11-15). 4. Genetic factors and the protective role of HLA molecules, that were present in our initial submission

We found risk in females and patients with chronic lower respiratory diseases, hypertension, and obesity comorbidities (Page 13, Line 2-10)

3.What's new in your article?

Response: Thank you for your comment. We mentioned the strengths of our study, including 1. This is the first systematic review of subject 2. We evaluated different domains concerning CD patients 3. We created two subgroups (based on the COVID-19 diagnosis method) to calculate the relative risk of COVID-19 acquisition 4. We included articles in higher levels of evidence and 45411 Celiac disease patients in the study 

(Page 13, Line 35-41)

4.PUT THESE CONCLUSIONS IN ABSTRACT: "females were more likely to be infected by COVID-19, and the

most common comorbidity in infected patients was a chronic lower respiratory disease;

around 10% of infected patients needed hospitalization, GFD adherence, and HR-QOL was

more or less the same before and during the pandemic, depression, anxiety, and stress levels

of patients varied among studies. Patients had more difficulties accessing GFPs based on

limited data

Response: Thank you for your comment. The conclusions mentioned above were put in the abstract (Page 2, Line 24-28)

5.DELETE THIS:

DELETE: "Further studies with larger populations are needed." BECAUSE The COVID19 pandemic is no more. If you ask for this, does it mean you want another COVID19 pandemic to come?

Response: Thank you for your comment. We deleted the sentence mentioned above; NO, ABSOLUTELY NOT; we do not wish for any pandemic or epidemy to come.

DELETE: "Further studies with more robust designs are needed to draw definite conclusions" - From what you have, show some conclusions!

Response: Thank you for your comment. We deleted the sentence mentioned above; and tried to show our conclusions!

---

## [Editor Report · Decision Letter 1]

4 May 2023

Celiac disease and COVID-19 in adults: A systematic review

PONE-D-23-03782R1

Dear Dr. Zarpoosh,

We’re pleased to inform you that your manuscript has been judged scientifically suitable for publication and will be formally accepted for publication once it meets all outstanding technical requirements.

Kind regards,

María de Lourdes Moreno Amador, Ph.D.

Academic Editor

PLOS ONE
---

## [Editor Report · Acceptance letter]

5 May 2023

PONE-D-23-03782R1 

Celiac disease and COVID-19 in adults: A systematic review 

Dear Dr. Zarpoosh:

I'm pleased to inform you that your manuscript has been deemed suitable for publication in PLOS ONE. Congratulations! Your manuscript is now with our production department. 

Kind regards, 

on behalf of

Dr. María de Lourdes Moreno Amador 

Academic Editor

PLOS ONE